# Transcriptional regulatory networks underlying gene expression changes in Huntington's disease

Seth A Ament[1,2,†], Jocelynn R Pearl[1,3,4,†], Jeffrey P Cantle[5], Robert M Bragg[5], Peter J Skene[6], Sydney R Coffey[5], Dani E Bergey[1], Vanessa C Wheeler[7], Marcy E MacDonald[7], Nitin S Baliga[1] (iD), Jim Rosinski[8], Leroy E Hood[1], Jeffrey B Carroll[5] (iD) & Nathan D Price[1,*] (iD)

## Abstract

Transcriptional changes occur presymptomatically and throughout Huntington's disease (HD), motivating the study of transcriptional regulatory networks (TRNs) in HD. We reconstructed a genome-scale model for the target genes of 718 transcription factors (TFs) in the mouse striatum by integrating a model of genomic binding sites with transcriptome profiling of striatal tissue from HD mouse models. We identified 48 differentially expressed TF-target gene modules associated with age- and CAG repeat length-dependent gene expression changes in *Htt* CAG knock-in mouse striatum and replicated many of these associations in independent transcriptomic and proteomic datasets. Thirteen of 48 of these predicted TF-target gene modules were also differentially expressed in striatal tissue from human disease. We experimentally validated a specific model prediction that SMAD3 regulates HD-related gene expression changes using chromatin immunoprecipitation and deep sequencing (ChIP-seq) of mouse striatum. We found CAG repeat length-dependent changes in the genomic occupancy of SMAD3 and confirmed our model's prediction that many SMAD3 target genes are downregulated early in HD.

**Keywords** Huntington's disease; SMAD3; transcription factor; transcriptional regulatory networks
**Subject Categories** Genome-Scale & Integrative Biology; Molecular Biology of Disease; Network Biology
**Mol Syst Biol. (2018) 14: e7435**

## Introduction

Massive changes in gene expression accompany many human diseases, yet we still know relatively little about how specific transcription factors (TFs) mediate these changes. Comprehensive characterization of disease-related transcriptional regulatory networks (TRNs) can help clarify potential disease mechanisms and prioritize targets for novel therapeutics. A variety of approaches have been developed to reconstruct interactions between TFs and their target genes, including models focused on reconstructing the physical locations of transcription factor binding (Gerstein *et al*, 2012; Neph *et al*, 2012), as well as computational algorithms utilizing gene co-expression to infer regulatory relationships (Friedman *et al*, 2000; Bonneau *et al*, 2006; Margolin *et al*, 2006; Huynh-Thu *et al*, 2010; Marbach *et al*, 2012; Reiss *et al*, 2015). These approaches have yielded insights into the regulation of a range of biological systems, yet accurate, genome-scale models of mammalian TRNs remain elusive.

Several lines of evidence point to a specific role for transcriptional regulatory changes in Huntington's disease (HD). HD is a fatal neurodegenerative disease caused by dominant inheritance of a polyglutamine (polyQ)-coding expanded trinucleotide (CAG) repeat in the *HTT* gene (MacDonald *et al*, 1993). Widespread transcriptional changes have been detected in post-mortem brain tissue from HD cases versus controls (Hodges *et al*, 2006), and transcriptional changes are among the earliest detectable phenotypes in HD mouse models (Luthi-Carter *et al*, 2000; Seredenina & Luthi-Carter, 2012; preprint: Bragg *et al*, 2016; Langfelder *et al*, 2016; Ament *et al*, 2017). These transcriptional changes are particularly prominent in the striatum, the most profoundly impacted brain region in HD (Vonsattel *et al*, 1985; Tabrizi *et al*, 2013). Replicable gene expression changes in the striatum of HD patients and HD mouse models include downregulation of genes related to synaptic function in medium spiny neurons accompanied by upregulation of genes

1   Institute for Systems Biology, Seattle, WA, USA
2   Institute for Genome Sciences and Department of Psychiatry, University of Maryland School of Medicine, Baltimore, MD, USA
3   Molecular & Cellular Biology Graduate Program, University of Washington, Seattle, WA, USA
4   Altius Institute for Biomedical Sciences, Seattle, WA, USA
5   Behavioral Neuroscience Program, Department of Psychology, Western Washington University, Bellingham, WA, USA
6   Basic Sciences Division, Fred Hutchinson Cancer Research Center, Seattle, WA, USA
7   Molecular Neurogenetics Unit, Center for Human Genetic Research, Department of Neurology, Massachusetts General Hospital, Harvard Medical School, Boston, MA, USA
8   CHDI Management, CHDI Foundation, Princeton, NJ, USA
    *Corresponding author. Tel: +1 206 732 1204; E-mail: nprice@systemsbiology.org
    †These authors contributed equally to this work

related to neuroinflammation (Seredenina & Luthi-Carter, 2012; Labadorf et al, 2015).

Some of these transcriptional changes may be directly related to the functions of the huntingtin (HTT) protein. Both wild-type and mutant HTT (mHTT) protein have been shown to associate with genomic DNA, and mHTT also interacts with histone-modifying enzymes and is associated with changes in chromatin states (Benn et al, 2008; Thomas et al, 2008; Seong et al, 2010). Wild-type HTT protein has been shown to regulate the activity of some TFs (Zuccato et al, 2007). Also, high concentrations of nuclear mHTT aggregates sequester TF and co-factor proteins and interfere with genomic target finding, though it is unknown whether this occurs at physiological concentrations of mHTT (Wheeler et al, 2000; Shirasaki et al, 2012; Li et al, 2016). Roles for several TFs in HD have been characterized (Zuccato et al, 2003; Arlotta et al, 2008; Tang et al, 2012; Dickey et al, 2015), but we lack a global model for the relationships between HD-related changes in the activity of specific TFs and the downstream pathological processes that they regulate.

The availability of large transcriptomics datasets related to HD is now making it possible to begin comprehensive network analysis of the disease, particularly in mouse models. Langfelder et al (2016) generated RNA-seq from the striatum of 144 knock-in mice heterozygous for an allelic series of HD mutations with differing CAG repeat lengths, as well as 64 wild-type littermate controls. They used gene co-expression networks to identify modules of co-expressed genes with altered expression in HD. However, their analyses did not attempt to identify any of the TFs responsible for these gene expression changes.

Here, we investigated the roles of core TFs that are predicted to drive the gene expression changes in HD, using a comprehensive network biology approach. We used a machine learning strategy to reconstruct a genome-scale model for TF-target gene interactions in the mouse striatum, combining publicly available DNase-seq with brain transcriptomics data from HD mouse models. We identified 48 core TFs whose predicted target genes were overrepresented among differentially expressed genes in at least five of fifteen conditions defined by a mouse's age and CAG repeat length, and we replicated the predicted core TFs and differential gene expression associations in multiple datasets from HD mouse models and from HD cases and controls. Based on the coordinated downregulation in HD knock-in mice of transcripts and proteins for *Smad3* and its predicted target genes, we hypothesized that SMAD3 may be a core regulator of early gene expression changes in HD. Using chromatin immunoprecipitation and deep sequencing (ChIP-seq), we demonstrate CAG repeat-dependent changes in SMAD3 occupancy and downregulation of SMAD3 target genes in mouse brain tissue. In conclusion, the results from our TRN analysis and ChIP-seq studies of HD reveal new insights into predicted transcription factor drivers of complex gene expression changes in this neurodegenerative disease.

# Results

## A genome-scale transcriptional regulatory network model of the mouse striatum

We reconstructed a model of TF-target gene interactions in the mouse striatum by integrating information about transcription factor

binding sites (TFBSs) with evidence from gene co-expression in the mouse striatum (Fig 1A). We predicted the binding sites for 871 TFs in the mouse genome using digital genomic footprinting. We identified footprints in DNase-seq data from 23 mouse tissues (Yue et al, 2014) using Wellington (Piper et al, 2013). Footprints are defined as short genomic regions with reduced accessibility to the DNase-I enzyme in at least one tissue. Our goal in combining DNase-seq data from multiple tissues was to reconstruct a single TFBS model that could make useful predictions about TF-target genes, even in conditions for which DNase-seq data were not available. We identified 3,242,454 DNase-I footprints. Genomic footprints are often indicative of occupancy by a DNA-binding protein. We scanned these footprints for 2,547 sequence motifs from TRANSFAC (Matys et al, 2006), JASPAR (Mathelier et al, 2014), UniProbe (Hume et al, 2015), and high-throughput SELEX (Jolma et al, 2013) to predict binding sites for specific TFs (TFBSs), and we compared these TFBSs to the locations of transcription start sites. We considered a TF to be a potential regulator of a gene if it had at least one binding site within a 5-kb region upstream and downstream of the TSS, which had been shown previously to maximize target gene prediction from digital genomic footprinting of the human genome (Plaisier et al, 2016).

To assess the accuracy of this TFBS model, we compared our TFBS predictions to ChIP-seq experiments from ENCODE (Yue et al, 2014) and ChEA (Lachmann et al, 2010; Appendix Fig S1). For 50 of 52 TFs, there was significant overlap between the sets of target genes predicted by our TFBS model versus ChIP-seq (FDR < 1%). Our TFBS model had a median 78% recall of target genes identified by ChIP-seq and a median 22% precision. That is, our model identified the majority of true-positive target genes but also made a large number of false-positive predictions. Low precision is expected in this model, since TFs typically occupy only a subset of their binding sites in a given tissue. Nonetheless, low precision indicates a need for additional filtering steps to identify target genes that are relevant in a specific context.

We sought to identify TF-target gene interactions that are active in the mouse striatum, by evaluating gene co-expression patterns in RNA-seq transcriptome profiles from the striatum of 208 mice (Langfelder et al, 2016). The general idea is that active regulation of a target gene by a TF is likely to be associated with strong TF-gene co-expression, and TFBSs allow us to identify direct regulatory interactions. This step also removes TFs with low expression: Of the 871 TFs with TFBS predictions, we retained as potential regulators the 718 TFs that were expressed in the striatum. We fit a regression model to predict the expression of each gene based on the combined expression patterns of TFs with one or more TFBSs ±5 kb of that gene's transcription start site. We used LASSO regularization to select the subset of TFs whose expression patterns together predicted the expression of the target gene. This approach extends several previous regression methods for TRN reconstruction (Tibshirani, 1996; Bonneau et al, 2006; Friedman et al, 2010; Chandrasekaran et al, 2011; Haury et al, 2012) by introducing TFBS-based constraints. In preliminary work, we considered a range of LASSO and elastic net ($\alpha$ = 0.2, 0.4, 0.6, 0.8, 1.0) regularization penalties and evaluated performance in fivefold cross-validation (see Materials and Methods). We selected LASSO based on the highest correlation between prediction accuracy in training versus test sets.

We validated the predictive accuracy of our TRN model by comparing predicted versus observed expression levels of each

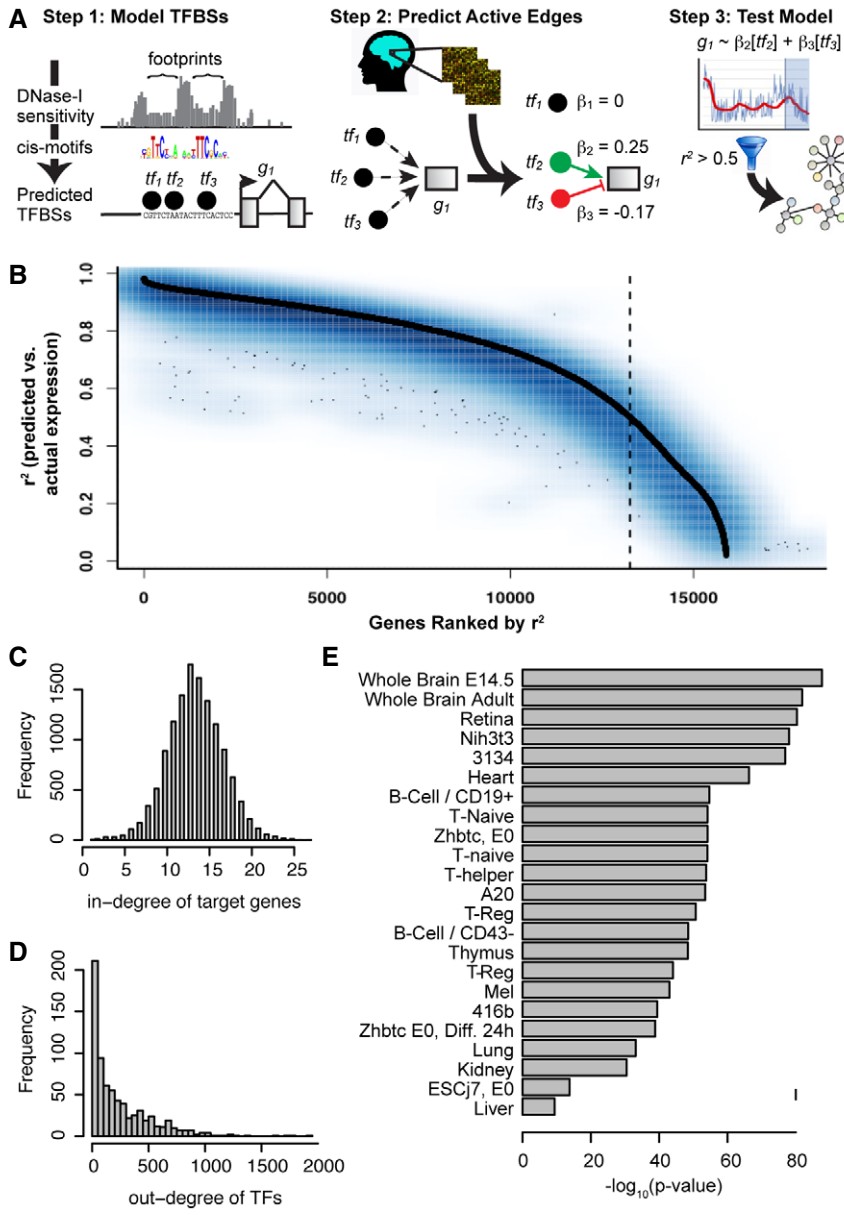

**Figure 1.  Reconstruction and validation of a transcriptional regulatory network (TRN) model of the mouse striatum.**

A   Schematic for reconstruction of tissue-specific TRN models by combining information about TF binding sites with evidence from co-expression.

B   Training (black) and test set (blue) prediction accuracy for genes in the mouse striatum TRN model. Genes are ordered on the x-axis according to their training set prediction accuracy ($r^2$, predicted versus actual expression). The dotted black line indicates the cut off for the number of genes which the model explained > 50% of expression variation in training data.

C   Distribution for the number of predicted regulators per target gene.

D   Distribution for the number of predicted target genes per TF.

E   Enrichments of TF-target gene interactions in the mouse striatum TRN for TFBSs supported by DNase footprints identified in 23 tissues.

gene. Our model explained > 50% of expression variation for 13,009 genes in training data (Fig 1B). Prediction accuracy in five-fold cross-validation was nearly identical to prediction accuracy in training data. That is, genes whose expression was accurately predicted in the training data were also accurately predicted in the test sets ($r = 0.94$; Fig 1B). Genes whose expression was not accurately predicted generally had low expression in the striatum (Appendix Fig S2). We removed poorly predicted genes, based on their training set accuracy before moving to the test set. The final TRN model contains 13,009 target genes regulated by 718 TFs via 176,518 interactions (Dataset EV1). Our model predicts a median of 14 TFs regulating each target gene and a median of 147 target genes per TF (Fig 1C and D). Fifteen TFs were predicted to regulate > 1,000 target genes (Appendix Fig S3). Importantly, TF-target gene interactions retained in our striatum-specific TRN model were enriched for genomic footprints in whole brains of 8-week-old

C57BL/6 male mice ($P$ = 1.4e-82) and in fetal brain ($P$ = 2.1e-88), supporting the idea that these TF-target gene interactions reflect TF binding sites in the brain (Fig 1E).

We defined "TF-target gene modules" as the sets of genes predicted to be direct targets of each of the 718 TFs. Of these 718 TF-target gene modules, 135 were enriched for a functional category from Gene Ontology (Ashburner *et al*, 2000; FDR < 5%, adjusting for 4,624 GO terms). Of the 718 TF modules, 337 were enriched ($P$ < 0.01) for genes expressed specifically in a major neuronal or non-neuronal striatal cell type (Doyle *et al*, 2008; Dougherty *et al*, 2010; Zhang *et al*, 2014), including known cell-type-specific activities for both neuronal (e.g., *Npas1-3*) and glia-specific TFs (e.g., *Olig1*, *Olig2*) (Appendix Fig S4). These results suggest that many TRN modules reflect the activities of TFs on biological processes within specific cell types.

## Prediction of core TFs associated with transcriptional changes in HD mouse models

We next sought to identify TFs that are core regulators of transcriptional changes in HD. Of the 208 mice in the RNA-seq dataset used for network reconstruction, 144 were heterozygous for a human *HTT* exon 1 allele knocked into the endogenous *Htt* locus (Wheeler *et al*, 1999; Menalled *et al*, 2003; Langfelder *et al*, 2016), and the remaining 64 mice were C57BL/6J littermate controls. Six distinct *Htt* alleles differing in the length of the CAG repeat were used. In humans, the shortest of these alleles—*Htt^{Q20}*—is non-pathogenic, and the remaining alleles—*Htt^{Q80}*, *Htt^{Q92}*, *Htt^{Q111}*, *Htt^{Q140}*, and *Htt^{Q175}*—are associated with progressively earlier onset of phenotypes. We used RNA-seq data generated by Langfelder *et al* (Langfelder *et al*, 2016) from four male and four female mice of each genotype at each of three time points: 2-month-old, 6-month-old, and 10-month-old mice. These mouse models undergo subtle age- and allele-dependent changes in behavior, and all of the ages profiled precede detectable neuronal cell death (Carty *et al*, 2015; Rothe *et al*, 2015; Alexandrov *et al*, 2016; preprint: Bragg *et al*, 2016).

We evaluated gene expression differences between *Htt^{Q20/+}* mice and mice with each of the five pathogenic *Htt* alleles at each time point, a total of 15 comparisons. The extent of gene expression changes increased in an age- and CAG length-dependent fashion, with extensive overlap between the DEGs identified in each condition (Fig 2). A total of 8,985 genes showed some evidence of differential expression (DEGs; $P$ < 0.01) in at least one of the 15 conditions, of which 5,132 were significant at a stringent false discovery rate < 1%. These results suggest that robust and replicable gene expression changes occur in the striatum of these HD mouse models at ages well before the onset of neuronal cell death or other overt pathology.

The predicted target genes of 209 TFs were overrepresented for DEGs in at least one of the 15 conditions (three ages × five mouse models; Fisher's exact test, $P$ < 1e-6; Dataset EV2). Repeating this analysis in 1,000 permuted datasets indicated that enrichments at this level of significance never occurred in more than four conditions (i.e., zero instances in 718,000 tests across 1,000 permutations of 718 TF-target gene networks). We therefore focused on a core set of 48 TFs whose predicted target genes were overrepresented for DEGs in five or more conditions. Notably, 44 of these 48 TFs were

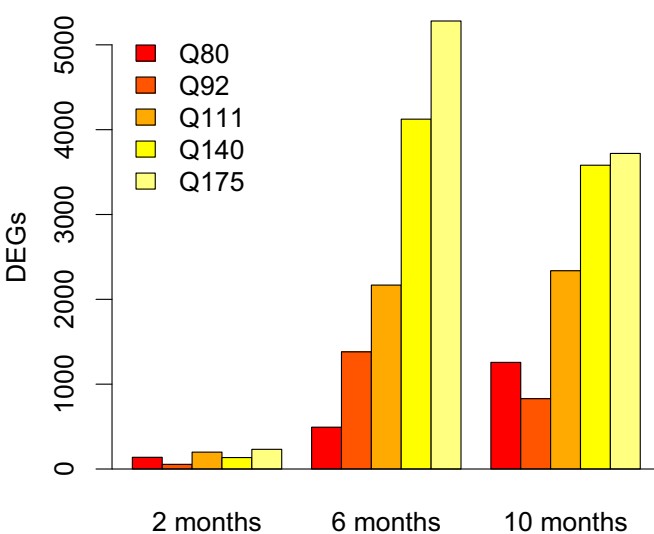

**Figure 2.  Robust changes in striatal gene expression in 2-, 6-, and 10-month-old HD knock-in mice.**

Counts of differentially expressed genes in each mouse model at each time point (allele shown versus Q20; edgeR log ratio test; nominal $P$-value < 0.01).

differentially expressed (FDR < 0.01) in at least one of the 15 conditions (Appendix Fig S5). We refer to these 48 TFs as core TFs.

## Replication of core TFs in independent datasets

We sought to replicate the associations of core TFs in HD by testing for enrichment of TF-target gene modules for differentially expressed genes or proteins in independent HD-related datasets. First, we conducted a meta-analysis of differentially expressed TF-target gene modules in four independent microarray gene expression profiling studies of striatal tissue from HD mouse models (Kuhn *et al*, 2007; Becanovic *et al*, 2010; Fossale *et al*, 2011; Giles *et al*, 2012). Targets of 46 of the 48 core TFs were enriched for DEGs (meta-analysis $P$-value < 0.01; Fig 3A and B) in the microarray data. The overlap between TFs whose target genes were differentially expressed in HD versus control mice in microarray datasets and the core TFs from our primary dataset was significantly greater than expected by chance (Fisher's exact test: $P$ = 5.7e-32). These results suggest that transcriptional changes in most of the core TF-target gene modules were preserved across multiple datasets and mouse models of HD.

Next, we asked whether the target genes of core TFs were also differentially abundant at the protein level. We studied quantitative proteomics data from the striatum of 64 6-month-old HD knock-in mice (Langfelder *et al*, 2016). These were a subset of the mice profiled with RNA-seq in our primary dataset. Targets of 22 of the 48 core TFs were enriched for differentially abundant proteins (Fisher's exact test, $P$ < 0.01; Fig 3A and B). The overlap between TFs whose target genes were differentially abundant between CAG-expanded versus wild-type mice and the core regulator TFs was significantly greater than expected by chance (Fisher's exact test: $P$ = 5.7e-20).

Third, we asked whether TFs predicted to drive early gene expression changes in mouse models of HD might also regulate gene expression changes in human disease. This analysis is complicated

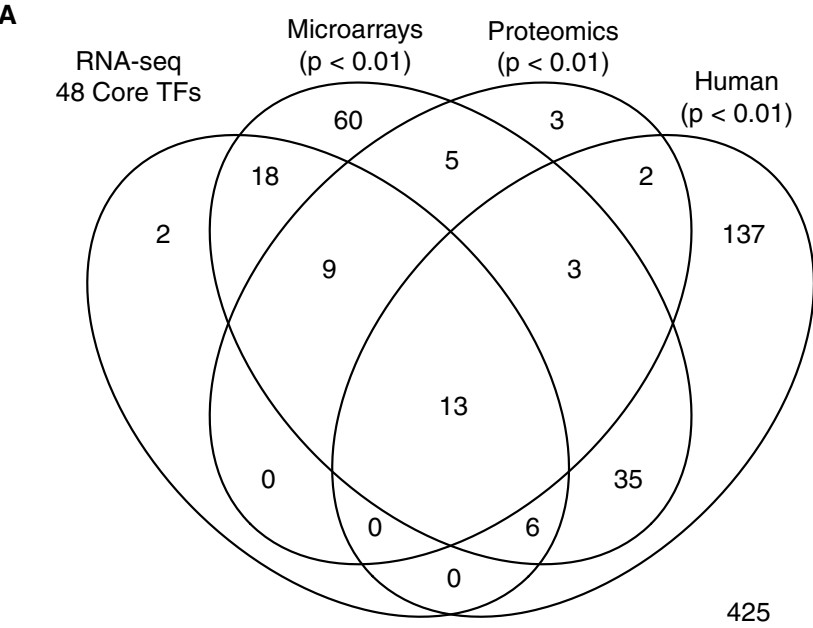

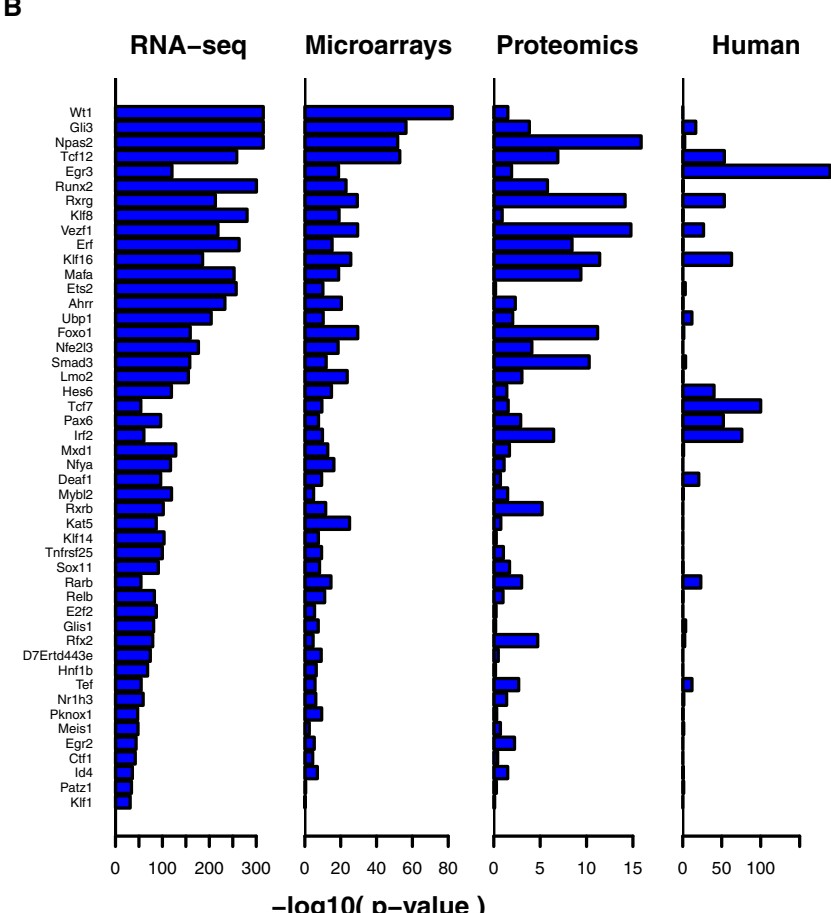

**Figure 3.   Replication of core TFs in independent datasets.**

A   Venn diagram showing overlap between core regulator TF-target gene modules identified in the primary RNA-seq dataset, compared to TF-target gene modules enriched for differentially expressed genes in three independent datasets.

B   $-\log_{10}(P\text{-values})$ for TF strength of enrichment of each of the core regulator TF-target gene modules for differentially expressed genes in each of the four datasets.

by the fact that striatal samples available from post-mortem HD patients are almost universally from late-stage disease, whereas our studies in mice focus on much earlier time points. In addition, the striatum is heavily degraded in late-stage HD, with many dead neurons and extensive astrogliosis (Vonsattel *et al*, 1985). For these reasons, transcriptomic changes in HD cases versus controls that are closely related to pathogenesis may be masked by a multitude of transcriptomic changes that are secondary to pathology. To overcome these issues and maximize our ability to detect overlap with the mouse models, we performed two tests in which we considered either a restrictive set of TFs from the HD mouse models (the 48 core regulators), as well as a broader set of TFs (all 209 TFs whose predicted target genes were enriched in at least one condition from our primary mouse RNA-seq dataset). We reconstructed a TRN model specific to the human striatum by integrating a map of TFBSs (Plaisier *et al*, 2016) based on digital genomic footprinting of 41 human cell types (Neph *et al*, 2012) with microarray gene expression profiles of post-mortem striatal tissue from 36 HD cases and 30 controls (Hodges *et al*, 2006). As in our TRN model for the mouse striatum, we fit a LASSO regression model to predict the expression of each gene in human striatum from the expression levels of TFs with predicted TFBSs within 5 kb of its transcription start sites (Appendix Fig S6). A total of 616 TFs had one-to-one orthology and ≥ 10 predicted target genes in both the mouse and human striatum TRN models. Using these 616 human TF-target gene modules, we tested the enrichment of differentially expressed genes in the caudate nucleus (part of the dorsal striatum) from HD cases versus controls (Hodges *et al*, 2006; Durrenberger *et al*, 2015). Predicted target genes for 13 of the 48 core TFs from mouse striatum were also overrepresented among differentially expressed genes in HD cases versus controls. This overlap was not statistically greater than expected by chance (odds ratio = 1.79; $P$ = 0.05; Fig 3A and B). However, when we considered the broader set of 209 TF-target gene modules from the primary mouse RNA-seq dataset, we found significant overlap for TF-target gene modules that were downregulated both in HD and in HD mouse models (28 shared TF-target gene modules; odds ratio = 3.6, $P$ = 5.0e-5; Appendix Fig S6D) and for TF-target gene modules that were upregulated both in HD and in HD mouse models (26 shared TF-target gene modules; odds ratio = 1.8, $P$ = 0.02; Appendix Fig S6E). These results suggest that some transcriptional programs are shared between the earliest stages of molecular progression (assayed in mouse models) and late stages of human disease. However, the human data support for relatively few of the core 48 TFs from mouse models.

Fourth, we asked whether core TFs in striatum also regulate HTT CAG length-dependent gene expression changes in other tissues. We analyzed gene expression in the cortex, hippocampus, cerebellum, and liver of HTT knock-in mice, using RNA-seq of these tissues from 168 of the mice in our primary striatal dataset (Langfelder *et al*, 2016). For each tissue, we reconstructed a transcriptional regulatory network model equivalent to our TRN model for mouse striatum, and we tested for the enrichment of Htt-allele-dependent gene expression changes among the predicted targets of each TF (Dataset EV3). We found a statistically significant overlap between the 48 core TFs in striatum versus the TF-target gene modules enriched for differentially expressed genes in each of the other four tissues (48 core TFs in striatum versus enriched modules in cortex: 16 shared TFs, odds ratio = 2.6, $P$ = 3.4e-3; striatum versus hippocampus: 21 shared TFs, odds ratio = 3.0, $P$ = 4.1e-4; striatum versus cerebellum: 17 shared TFs, odds ratio = 2.17, $P$ = 1.3e-2; striatum versus liver: 25 shared TFs, odds ratio = 3.3, $P$ = 8.2e-5). These analyses revealed a wide range of tissue specificity for the associations of the 48 core striatal TFs with HTT CAG length-dependent gene expression changes (Appendix Fig S7). For instance, the predicted targets of RXRG were enriched for differentially expressed genes in all five tissues, whereas targets of IRF2 were enriched only in striatum.

Notably, targets of 13 of the 48 core regulator TFs were enriched for differentially expressed genes in all four striatal datasets: GLI3, IRF2, KLF16, NPAS2, PAX6, RARB, RFX2, RXRG, SMAD3, TCF12, TEF, UBP1, and VEZF1. These 13 TFs may be especially interesting for follow-up studies.

## Biological associations of core TFs

We evaluated relationships among the 48 core TFs based on clustering and network topology. Plotting TF-to-TF regulatory interactions among the 48 core TFs (Fig 4A–D) revealed two distinct TF-to-TF sub-networks, characterized by numerous positive interactions within sub-networks and by fewer, mostly inhibitory interactions between sub-networks. The target genes of TFs in the first sub-network were predominantly downregulated in HD, while the target genes of TFs in the second sub-network were predominantly upregulated. Hierarchical clustering of the 48 core TFs based on the expression patterns of their predicted target genes revealed similar groupings of TFs whose target genes were predominantly down- versus upregulated (Fig 5).

We studied the predicted target genes of each core TF to characterize possible roles for these TFs in HD. Downregulated TF-target gene modules were overrepresented for genes specifically expressed in DRD1$^+$ and DRD2$^+$ medium spiny neurons (Fig 5). Functional enrichments within these modules were mostly related to synaptic function, including metal ion transmembrane transporters (targets of NPAS2, $P$ = 2.3e-4), voltage-gated ion channels (targets of MAFA, $P$ = 8.1e-4), and protein localization to cell surface (targets of RXRG, $P$ = 1.7e-4). These network changes may be linked to synapse loss in medium spiny neurons, which is known to occur in knock-in mouse models of HD (Deng *et al*, 2013).

Some upregulated TF-target gene modules were overrepresented for genes specifically expressed in oligodendrocytes or astrocytes, while others were overrepresented for genes specifically expressed in neurons (Fig 5). Functional enrichments within these modules included Gene Ontology terms related to apoptosis ("positive regulation of extrinsic apoptotic signaling pathway via death domain receptors," targets of WT1, $P$ = 1.8e-4) and DNA repair (targets of RUNX2, "single-strand selective uracil DNA N-glycosylase activity," $P$ = 2.0e-4). Therefore, core TFs whose target genes were predominantly upregulated may contribute to a variety of pathological processes both in neurons and in glia. Oligodendrocyte counts have been shown to be increased in HD mutation carriers, whereas micro- and astrogliosis are thought to begin later in disease progression (Vonsattel *et al*, 1985).

An open question in the field is whether the same sequence of pathogenic events underlies disease progression in juvenile-onset HD due to *HTT* alleles with CAG tracts with > 60 repeats versus adult-onset HD due to *HTT* alleles with ∼ 40–60 CAG repeats (Nance & Myers, 2001). This question is of practical relevance for

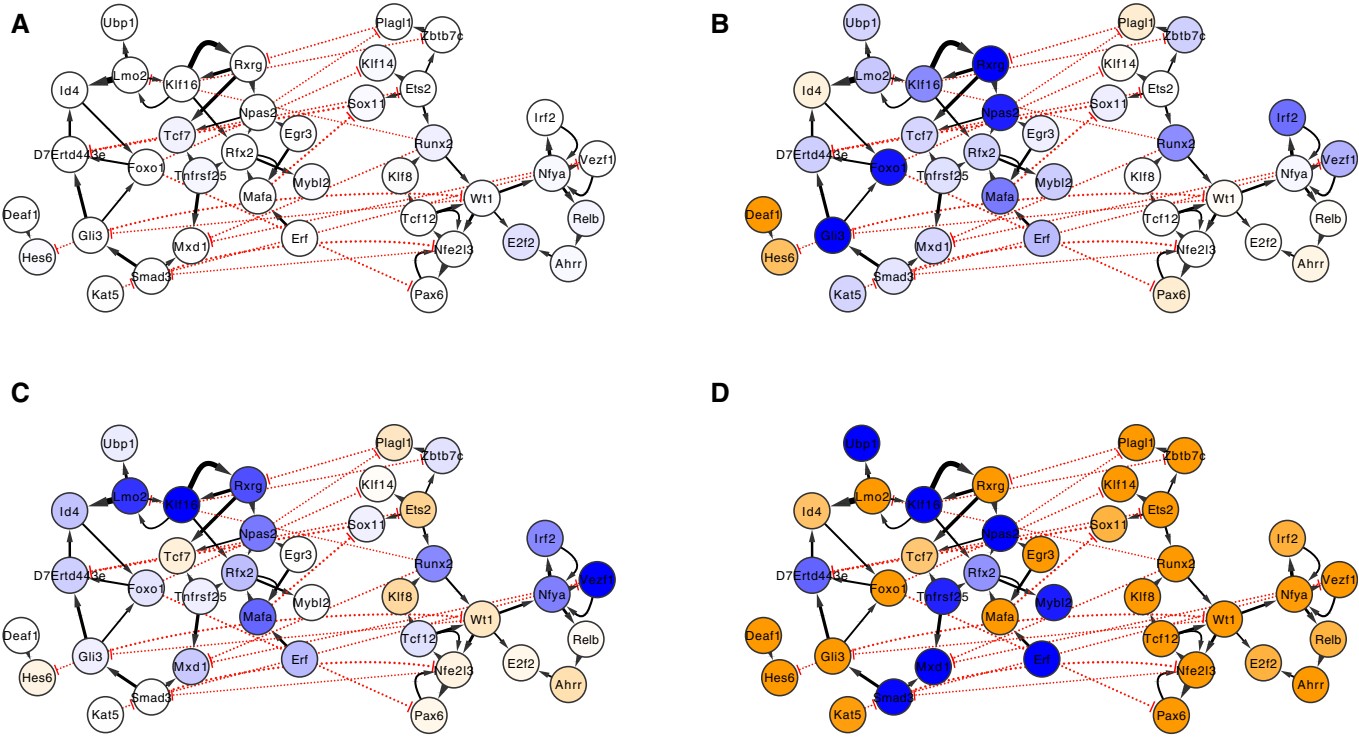

**Figure 4.  Predicted TF-to-TF interactions among 48 putative core regulators of transcriptional changes in mouse models of Huntington's disease.**

A–D    Nodes and edges indicate direct regulatory interactions between TFs predicted by the mouse striatum TRN model. Solid black arrows and dotted red arrows indicate positive versus inhibitory regulation, respectively, and the width of the line is proportional to the predicted effect size. Blue and orange shading of nodes indicates that the TF's target genes are overrepresented for downregulated versus upregulated genes in HD mouse models, respectively. If a TF's target genes are enriched in both directions, the stronger enrichment is shown. Each panel indicates the network state in a specific condition: (A) 2-month-old $Htt^{Q92/+}$ mice, (B) 6-month-old $Htt^{Q92/+}$ mice, (C) 2-month-old $Htt^{Q175/+}$ mice, or (D) 6-month-old $Htt^{Q175/+}$ mice.

modeling HD in mice, since mouse models with very long CAG tracts are often used in research due to their faster rates of phenotypic progression within a 2-year lifespan. To address this question, we evaluated overlap between TF-target gene modules activated at the earliest time points in mice with each of the five pathogenic $Htt$ alleles in our dataset. In the mice with the longest $Htt$ alleles—$Htt^{Q175}$ and $Htt^{Q140}$—the target genes of core TFs first became enriched for differentially expressed genes in 2-month-old mice. In mice with relatively short $Htt$ alleles—$Htt^{Q111}$, $Htt^{Q92}$, and $Htt^{Q80}$—target genes of core TFs became enriched for differentially expressed genes beginning in 6-month-old mice. We found that eight modules—the predicted target genes of IRF2, MAFA, KLF16, LMO2, NPAS2, RUNX2, RXRG, and VEZF1—were significantly enriched for DEGs in at least three of these five conditions (2-month-old $Htt^{Q175/+}$, 2-month-old $Htt^{Q140/+}$, 6-month-old $Htt^{Q111/+}$, 6-month-old $Htt^{Q92/+}$, and 6-month-old $Htt^{Q80/+}$). A limitation of this analysis is that the alleles used in this study are associated with juvenile-onset disease, and the extent to which these results extend to adult-onset alleles remains to be determined. Nonetheless, these results suggest that many aspects of the trajectory of transcriptional changes are shared across the CAG lengths that have been studied. Notably, all the TFs whose target genes were enriched for differentially expressed genes at the very earliest time points were enriched primarily for genes that were downregulated in HD. Strong enrichments of TF-target gene modules for upregulated genes occurred only at slightly later time points.

## Genome-wide characterization of SMAD3 binding sites in the mouse striatum supports a role in early gene dysregulation in HD

We selected the TF SMAD3 for functional validation for the following reasons. SMAD3 was one of 13 core TFs whose predicted target genes were overrepresented among differentially expressed genes across all four independent datasets. SMAD3's predicted target genes were predominantly downregulated in an age- and CAG length-dependent fashion, beginning at or before 6 months of age (Fig 5). SMAD3 acts primarily downstream of TGF-β signaling, making it a potential drug target. In addition, an initial screen of antibodies to several of the core TFs revealed a high-quality SMAD3 antibody, suitable for chromatin immunoprecipitation.

Decreased expression of SMAD3 target genes could result from a change in SMAD3 expression. In addition, changes in the expression levels of SMAD3 target genes could result from a change in TGF-β signaling, as SMAD3 activation and nuclear localization depend on its phosphorylation at Ser423 and Ser425 by the TGF-β receptor (Liu *et al*, 1997). To evaluate these possibilities, we examined *Smad3* RNA, phospho-Ser423/425-SMAD3 protein, and total SMAD3 protein in the striatum of HD knock-in mice versus wild-type controls. We detected an age- and CAG length-dependent decrease in *Smad3* RNA, similar to the expression of its predicted target genes (Fig 6A). In addition, Western blots revealed a trend toward a lower ratio of phospho-Ser423/425-SMAD3 to total SMAD3 in the striatum

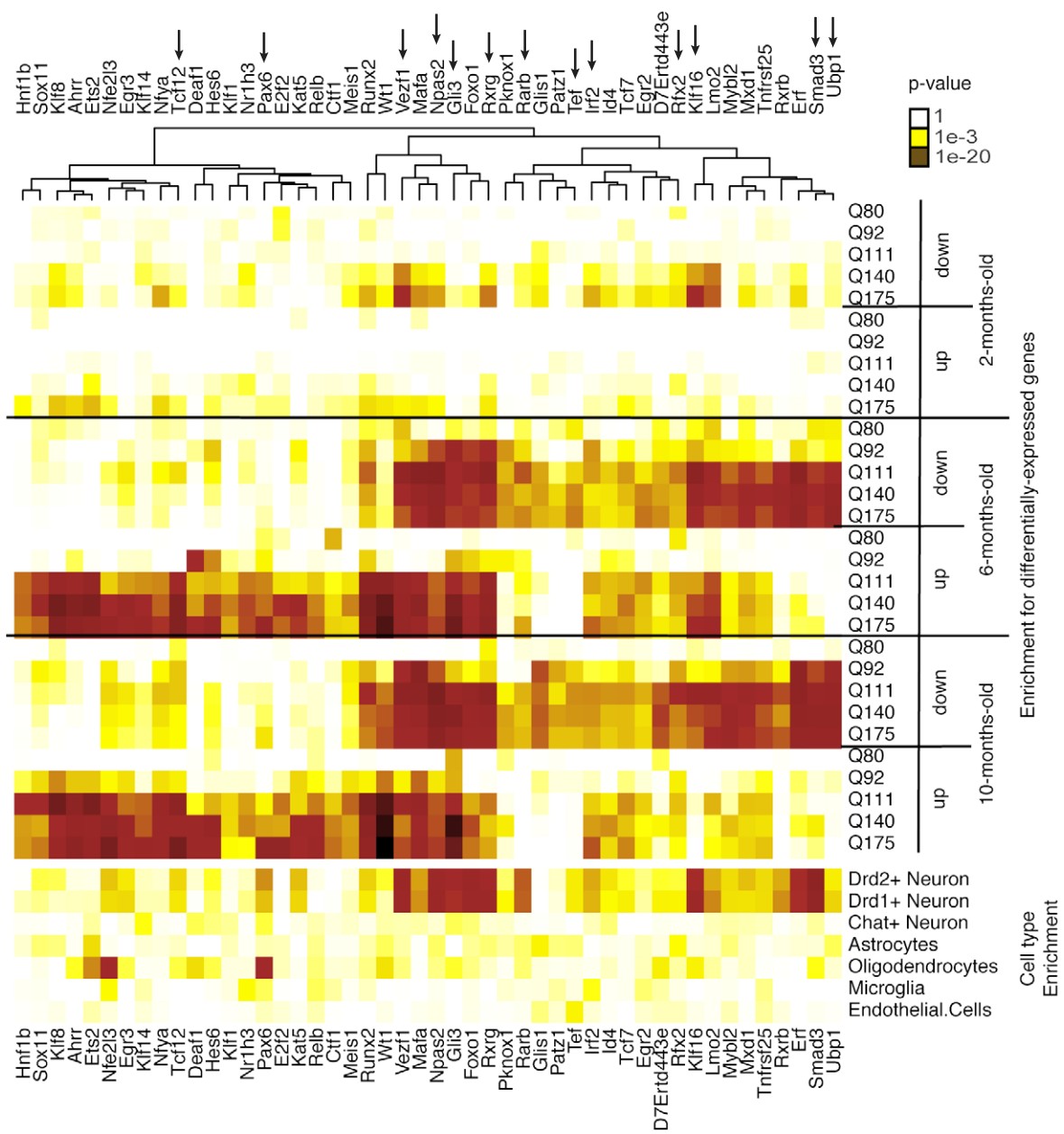

**Figure 5.   Enrichments of the 48 core TFs for differentially expressed genes in each condition and for cell-type-specific genes.**

Heatmap showing the enrichments of each TF's target genes for down- and upregulated genes for each *Htt* allele at each time point as well as enrichments of each TF's target genes for genes expressed specifically in one of seven major cell types in the mouse striatum. Arrows at top indicate the 13 TFs with replication in all four independent datasets.

of 4- and 11-month-old $Htt^{Q111/+}$ mice, compared to wild-type controls (ANOVA, genotype: $F_{1,16} = 3.714$, $P = 0.072$; age: $F_{1,16} = 4.590$, $P = 0.048$; interaction: $F_{1,16} = 0.304$, $P = 0.589$), suggesting a possible decrease in the activation by TGF-β (Appendix Fig S8). By contrast, we did not detect a significant change in total SMAD3 protein in these mice (ANOVA, genotype: $F_{1,16} = 0.487$, $P = 0.495$; age: $F_{1,16} = 0.506$, $P = 0.487$; interaction: $F_{1,16} = 1.085$, $P = 0.313$; Appendix Fig S8). Similarly, quantitative proteomics of an allelic series of 6-month-old HD knock-in mice revealed a non-significant trend toward decreased total SMAD3 protein (Pearson's correlation;

SMAD3 versus *Htt* CAG length: $r = -0.25$, $P$-value = 0.12). In summary, we find evidence for decreased *Smad3* RNA expression and a trend toward decreased SMAD3 activation by TGF-β in the striatum of HD knock-in mice, though any changes in SMAD3 protein are subtle. Overall, these results support our prediction from network modeling that decreased SMAD3 activity is an early event in the striatum of *Htt* CAG knock-in mice.

Next, we characterized the binding sites of SMAD3 in the striatum of 4-month-old $Htt^{Q111/+}$ mice and wild-type littermate controls to validate and extend our network predictions. We performed

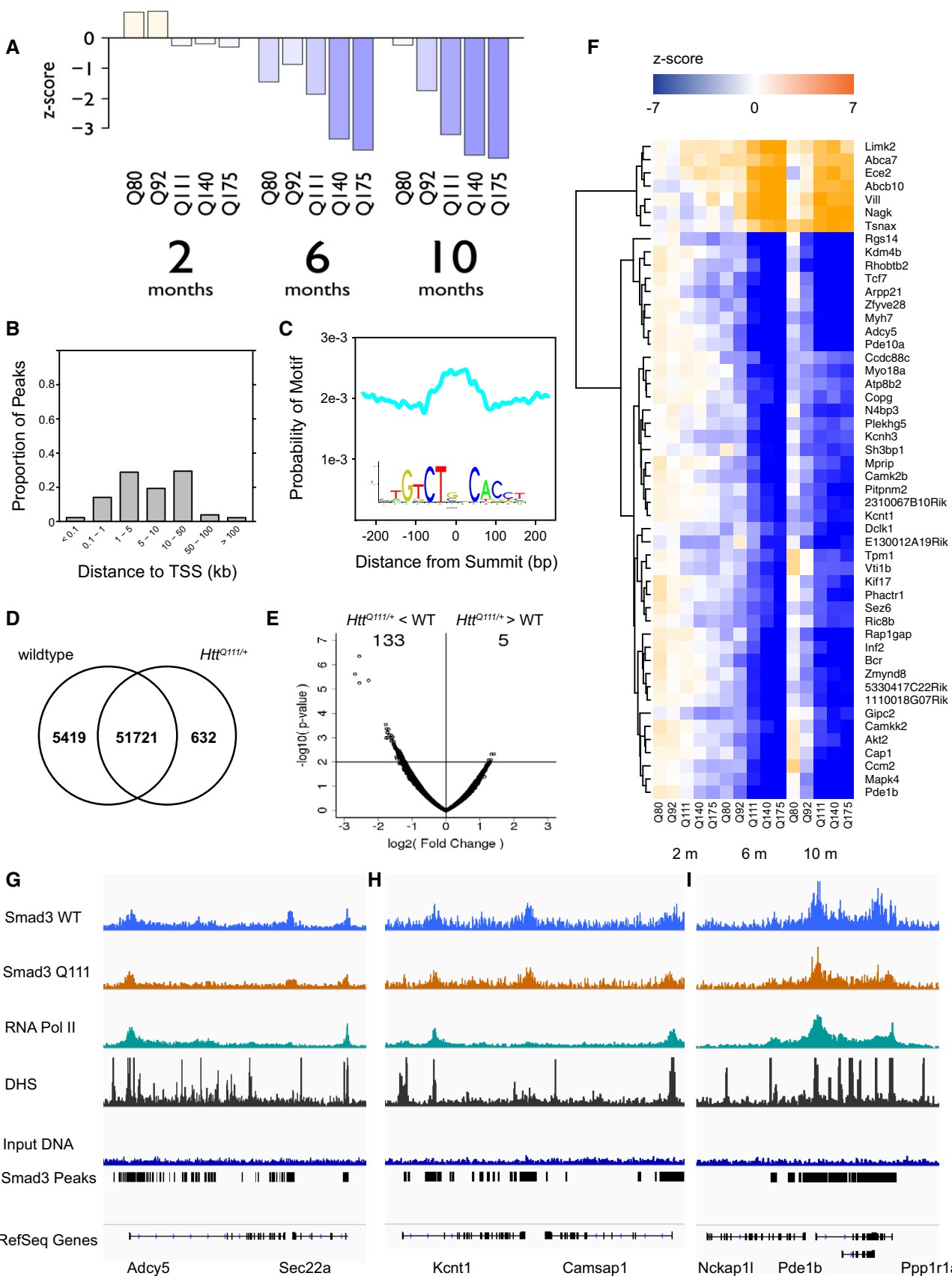

**Figure 6.**

**Figure 6.  SMAD3 expression, genomic occupancy, and target gene expression in the striatum of HD mouse models.**

A  Progressive age- and *Htt*-allele-dependent changes in the expression of SMAD3 in mouse striatum. Bars indicate *z*-scores for the expression level in heterozygous mice with each pathogenic *Htt* allele compared to age-matched *Htt*$^{Q20/+}$ mice.

B  Distribution of the distances of 57,772 SMAD3 peaks identified by ChIP-seq to the nearest transcription start site (TSS).

C  The summits of SMAD3 peaks are enriched for the sequence motif recognized by SMAD3 (JASPAR CORE MA0513.1, shown in inset).

D  Overlap between peaks identified in *Htt*$^{Q111/+}$ versus wild-type mice.

E  SMAD3 occupancy is decreased at a subset of peaks in *Htt*$^{Q111/+}$ versus wild-type mice. *x*-axis and *y*-axis represent the log2(fold change) and −log10(*P*-value), respectively, for each peak region.

F  Age- and *Htt*-allele-dependent expression patterns of the top 50 most strongly differentially expressed SMAD3 target genes.

G–I  Genomic occupancy of SMAD3 and RNA polymerase II and accessibility of genomic DNA to DNase-I near (G) *Adcy5*, (H) *Kcnt1*, and (I) *Pde1b*.

chromatin immunoprecipitation and deep sequencing using an antibody specific to SMAD3 (ChIP-seq; $n = 2$ pooled samples per group, with each pool containing DNA from three mice). Peak-calling revealed 57,772 SMAD3 peaks (MACS2.1, FDR < 0.01 and > 10 reads in at least two of the four samples). Of the 57,772 SMAD3 peaks, 34,633 (59.9%) were located within 10 kb of transcription start sites (TSSs), including at least one peak within 10 kb of the TSSs for 11,727 genes (Fig 6B). The summits of SMAD3 peaks were enriched for the SMAD2:SMAD3:SMAD4 motif (*P*-value = 7.2e-85; Fig 6C). Importantly, the TSSs for 753 of the 938 computationally predicted SMAD3 target genes in our TRN model were located within 10 kb of at least one ChIP-based SMAD3 binding site. This overlap was significantly greater than expected by chance (odds ratio = 4.33, *P*-value = 2.8e-84).

We characterized the relationship between SMAD3 occupancy and transcriptional activation by measuring the genomic occupancy of RNA polymerase II (RNAPII) in the striatum of *Htt*$^{Q111/+}$ and wild-type mice. RNAPII occupancy is a marker of active transcription and of active transcription start sites. Occupancy of SMAD3 and occupancy of RNAPII were positively correlated, across all genomic regions ($r = 0.70$) and specifically within SMAD3 peaks ($r = 0.71$). Thus, SMAD3 binding is associated with active transcription.

Similarly, we characterized the relationship between SMAD3 occupancy and chromatin accessibility, using publicly available DNase-seq of midbrain tissue from wild-type mice. Of the 57,772 SMAD3 peaks, 22,650 (39.2%) overlapped a DNase-hypersensitive site in the midbrain. Occupancy of SMAD3 was positively correlated with DNase-I hypersensitivity across all genomic regions ($r = 0.33$) and specifically within SMAD3 peaks ($r = 0.25$). Thus, SMAD3 binding sites are enriched for signatures of active enhancers.

We ranked genes from highest to lowest SMAD3 regulatory potential based on the number of SMAD3 peaks within 10 kb of their transcriptional start sites. We focused on the top 837 genes with SMAD3 peak counts > 2 standard deviations above the mean. These top 837 SMAD3 target genes were enriched (FDR < 0.01) for 24 non-overlapping clusters of Gene Ontology terms (Appendix Table S1). These enriched GO terms prominently featured pathways related to gene regulation ("mRNA processing", $P = 4.2e-9$; "histone modification", $P = 1.7e-7$; "transcriptional repressor complex", $P = 3.7e-5$), as well as functions more specifically related to brain function ("neuromuscular process controlling balance", $P = 1.2e-7$; "brain development", $P = 1.27e-6$; "neuronal cell body", $P = 2.5e-5$).

We performed quantitative and qualitative analyses to compare SMAD3 occupancy in *Htt*$^{Q111/+}$ versus wild-type mice. Of the 57,772 SMAD3 peaks, 51,721 (89.5%) were identified in both *Htt*$^{Q111/+}$ and wild-type mice. A total of 5,419 peaks (9.4%) were

identified only in wild-type mice, while only 632 peaks (1.1%) were identified only in *Htt*$^{Q111/+}$ mice (Fig 6D). Quantitative analyses of differential binding with edgeR revealed four peaks whose occupancy was significantly different (FDR < 0.05) between *Htt*$^{Q111/+}$ and wild-type mice. All four of these peaks were more weakly occupied in *Htt*$^{Q111/+}$ mice. A total of 138 peaks had nominally significant differences in occupancy between genotypes ($P < 0.01$). Of these 138 peaks, 133 (96.4%) were more weakly occupied in *Htt*$^{Q111/+}$ mice (Fig 6E). These results suggest that SMAD3 occupancy is decreased at a subset of its binding sites in 4-month-old *Htt*$^{Q111/+}$ mice.

Finally, we tested whether the top 837 SMAD3 target genes from ChIP-seq were differentially expressed in HD knock-in mice. The top 837 SMAD3 target genes from ChIP-seq were significantly overrepresented among genes that became downregulated in the striatum of HD knock-in mice (223 downregulated SMAD3 target genes; odds ratio = 2.0, *P*-value = 3.4e-15; Fig 6F). Example target gene tracks are shown in Fig 6G–I including differentially expressed genes *Adcy5, Kcnt1,* and *Pde1b*. By contrast, SMAD3 target genes were not overrepresented among genes that became upregulated in the striatum of HD mouse models (143 upregulated SMAD3 target genes, odds ratio = 0.92, $P = 0.40$). These results are consistent with our computational model, in which SMAD3 target genes were primarily downregulated in HD knock-in mice. Therefore, reduced SMAD3 binding is associated with downregulation of its target genes in HD mouse models.

## Discussion

Here, we identified putative core TFs regulating gene expression changes in HD by reconstructing genome-scale transcriptional regulatory network models for the mouse and human striatum. Identifying core TFs in HD provides insights into the mechanisms of this devastating, incurable disease. This method to reconstruct models of mammalian transcriptional regulatory networks can be readily applied to find regulators underlying any trait of interest.

Our model extends prior knowledge about the TFs involved in HD. A role in HD for RARB is supported by ChIP-seq and transcriptome profiling of striatal tissue from *Rarb*$^{-/-}$ mice (Niewiadomska-Cimicka *et al*, 2016). A role in HD for FOXO1 is supported by experimental evidence that FOXO signaling influences the vulnerability of striatal neurons to mutant huntingtin (Parker *et al*, 2012). A role in HD for RELB is supported by experimental evidence that NF-κB signaling mediates aberrant neuroinflammatory responses in HD and HD mouse models (Hsiao *et al*, 2013). Notably, microglia counts in 10–12 months *Htt*$^{Q111/+}$ mice indicate that these cells are

not proliferating (preprint: Bragg *et al*, 2016), suggesting that the transcriptional changes observed in our study represent a proinflammatory state, rather than microgliosis *per se*. Other predicted core TFs, including KLF16 and RXRG, have previously been noted among differentially expressed genes in mouse models of HD (Seredenina & Luthi-Carter, 2012). In some cases, known functions for core TFs suggest hypotheses about their roles in HD. For instance, NPAS2 is a component of the molecular clock, so its dysfunction could contribute to circadian disturbances in HD (Morton *et al*, 2005). Notably, the predicted target genes for several TFs whose functions in HD have been studied by other investigators —for example, REST (Zuccato *et al*, 2003), SREBF2 (Valenza *et al*, 2005), and FOXP1 (Tang *et al*, 2012)—were overrepresented for differentially expressed genes in our model, but only at later time points or more weakly than our top 48 core regulator TFs.

Our results suggest that HD involves parallel, asynchronous changes in distinct down- versus upregulated TF sub-networks. Targets of TFs in the downregulated sub-network are enriched for synaptic genes and appear to be primarily neuronal. Targets of TFs in the upregulated sub-network are enriched for stress response pathways (e.g., DNA damage repair, apoptosis). These upregulated networks appear to involve processes occurring in both neurons and glia. Downregulation of synaptic gene networks preceded upregulation of stress response gene networks, suggesting that the synaptic changes are more proximal to the mutant HTT protein. A large body of prior work provides independent support for synaptic changes in medium spiny neurons and of activated gliosis in HD pathogenesis (Singhrao *et al*, 1999; Deng *et al*, 2013; Hsiao *et al*, 2013).

Replication across four independent datasets revealed 13 TFs whose target genes were most consistently enriched among differentially expressed genes. We propose that these TFs should be prioritized for follow-up experiments, both to validate predicted target genes and to evaluate specific biological functions for each TF. For instance, it will be interesting to determine which (if any) of the core TFs have direct protein–protein interactions with the HTT protein and to test our model's predictions about TF perturbations with specific aspects of HD pathology. The target genes for most of these 13 TFs were enriched for genes that were downregulated in HD and for neuron-specific genes, consistent with the idea that pathological changes originate in medium spiny neurons. It is important to note that independent datasets comprised from different mouse models, ages, and data collection centers might dilute the reproducibility of key comparisons. We feel that our analysis approach—comparing across multiple independent studies—is therefore more stringent and retains only those predictions for which there is consistent reproducibility.

Network modeling of SMAD3 target genes, changes in SMAD3 expression and phosphorylation, and SMAD3 ChIP-seq suggest that SMAD3 and its target genes are downregulated in the striatum of HD knock-in mice. Previous studies have described changes in SMADs and upstream components of the TGF-β signaling pathway in cellular and mouse models of HD, as well as in blood from HD cases versus controls, but the direction of these effects was contradictory between studies (Battaglia *et al*, 2011; Ring *et al*, 2015; Bowles *et al*, 2017). Our results are the first characterization of this system in the striatum of a genetically accurate mouse model with physiological expression of mutant HTT and provide the first evidence linking TGF-β and SMAD3 to downstream transcriptomic changes in HD mouse models. These findings suggest an intriguing

possibility that agonists of TGF-β signaling could have therapeutic benefit in HD patients. Consistent with this possibility, TGF-β treatment reduced apoptotic cell death in neural stem cells with expanded HTT CAG tracts (Ring *et al*, 2015).

Our method to reconstruct TRNs by integrating information about TF occupancy with gene co-expression is likely to be broadly applicable, providing a strategy to optimize both mechanistic and quantitative accuracy. TRN reconstruction methods are based purely on gene co-expression struggle to distinguish direct versus indirect interactions. Physical models of TF occupancy provide poor quantitative predictions because many TF binding sites are non-functional or do not regulate the nearest gene. Our study demonstrates that integrated TRN modeling can be utilized effectively to study neurodegenerative diseases such as HD, combining data from the ENCODE project with disease-specific transcriptome profiling.

# Materials and Methods

### Referenced datasets

We obtained RNA-seq and microarray gene expression profiling data from the following GEO Datasets (http://www.ncbi.nlm. nih.gov/geo/): GSE65776 (Langfelder *et al*, 2016), GSE73508, GSE18551 (Becanovic *et al*, 2010), GSE32417 (Giles *et al*, 2012), GSE9038 (Fossale *et al*, 2011), GSE9857 (Kuhn *et al*, 2007), GSE26927 (Durrenberger *et al*, 2015), and GSE3790 (Hodges *et al*, 2006). We obtained proteomics data from the PRIDE archive (https://www.ebi.ac.uk/pride/archive/), accession PXD003442 (Langfelder *et al*, 2016). For RNA-seq data (GSE65776), we downloaded read counts and FPKM estimates, mapped to ENSEMBL gene models. For Affymetrix microarrays (GSE18551, GSE32417, GSE9038, GSE9857, GSE26927, and GSE3790), we downloaded raw image files and used the affy package in R to perform within-sample RMA normalization and between-sample quantile normalization. For proteomics data, we downloaded MaxQuant protein quantities.

### Genomic footprinting

DNase-I digestion of genomic DNA followed by deep sequencing (DNase-seq) enables the identification of genomic footprints across the complete genome. We predicted genome-wide transcription factor binding sites (TFBSs) in the mouse and human genomes based on instances of TF sequence motifs in digital genomic footprints from the ENCODE project. Short regions of genomic DNA occupied by DNA-binding proteins produce characteristic "footprints" with altered sensitivity to the DNase-I enzyme. DNase-I digestion of genomic DNA followed by deep sequencing (DNase-seq) enables the identification of genomic footprints across the complete genome.

For the human TFBS model, we used a previously described database (Plaisier *et al*, 2016) of footprints from DNase-seq of 41 cell types (Neph *et al*, 2012). For the mouse TFBS model, we downloaded digital genomic footprinting data (deep DNase-seq) for 23 mouse tissues and cell types (Yue *et al*, 2014) from the UCSC ENCODE portal on October 29, 2013: ftp://hgdownload.cse.ucsc.edu/goldenPath/mm9/database/. We detected footprints in each sample with Wellington (Piper *et al*, 2013), using a significance threshold, $P < 1e\text{-}10$. Using FIMO (Grant *et al*, 2011), we scanned

the mouse genome (mm9) for instances of 2,547 motifs from TRANSFAC (Matys *et al*, 2006), JASPAR (Mathelier *et al*, 2014), UniPROBE (Hume *et al*, 2015), and high-throughput SELEX (Jolma *et al*, 2013). We intersected footprints from all tissues with motif instances to generate a genome-wide map of predicted TFBSs. A motif can be recognized by multiple TFs with similar DNA-binding domains. We assigned motifs to TF families using annotations from the TFClass database (Wingender *et al*, 2013). In total, our model included motifs recognized by 871 TFs.

**Regression-based transcriptional regulatory network models**

We fit a regression model to predict the expression of each gene in mouse striatum, cortex, hippocampus, cerebellum, and liver, as well as in human striatum, based on the expression patterns of TFs that had predicted binding sites within 5 kb of that gene's transcription start sites. We applied LASSO regularization to penalize regression coefficients and remove TFs with weak effects, using the glmnet package in R. These methods were optimized across several large transcriptomics datasets, prior to their application to the Huntington's disease data. To reconstruct the TRN model for mouse striatum, we used RNA-seq data from the striatum of 208 mice (Langfelder *et al*, 2016). Prior to network reconstruction, we evaluated within- and between-group variance and detected outlier samples using hierarchical clustering and multidimensional scaling. No major differences in variance were identified between groups, and no outlier samples were detected or removed.

We considered a variety of model parameterization during the initial model formulation. We considered elastic net regression and ridge regression as alternatives to LASSO regression. We selected LASSO based on the least falloff in performance from the training data to test sets in fivefold cross-validation. We note that when multiple TFs have correlated expression, the LASSO will generally retain only one for the final model. This feature of the LASSO has been considered advantageous, since it can eliminate indirect interactions. However, this feature also has a downside in that there is virtually no doubt that the TFs selected by our model underestimate the true number of TF-target gene interactions. We would only pick up dominant effects where a linear model works reasonably well. Our primary interest is ultimately in using this approach to find a relatively small number of targets based on multiple lines of evidence. We are less concerned here with finding everything than in trying to make sure what we do find is as highly enriched for true positives as possible.

We also considered a variety of strategies to select an appropriate penalty parameter. For instance, we could apply an independent penalty parameter for each gene, or we could use a uniform penalty parameter across all genes. We found that optimal performance was obtained in both training data and fivefold cross-validation when we applied a uniform penalty parameter across all genes. We assigned this penalty parameter by evaluating performance in cross-validation across a range of possible parameters for a random subset of 100 genes. For each gene, we identified the most stringent penalty such that the unfitted variance was < 1 standard error greater than the minimum unfitted variance across all the penalty parameters considered. We selected the median penalty defined by this procedure across the 100 randomly selected gene.

Not all genes' expression can be accurately predicted based on the expression of TFs. To select genes for the final model, we evaluated the variance explained by the model in a training set consisting of 80% of the data. We selected those genes for which the model explained > 50% of expression variance in the training set and carried these genes forward to a test set, consisting of the remaining 20% of genes. We found that training set performance accurately predicted test performance ($r = 0.94$). We therefore fit a final model for genes whose expression could be accurately predicted in the training set. The result of these procedures is a tissue-specific TRN model, predicting the TFs that regulate each gene in the striatum and assigning a positive or negative weight for each TF's effect on that gene's expression in the striatum.

**Enrichments of TF-target gene modules in ChIP-seq data**

We downloaded ChIP-seq data from the ENCODE website (encode-project.org, accessed on August 20, 2015) for 33 mouse transcription factors included in our TRN model. We identified genes whose transcription start sites were located within 5 kb of a narrowPeak in each ChIP experiment. We also downloaded a table of ChIP-to-gene annotations for 19 additional mouse TFs from the ChEA website (http://amp.pharm.mssm.edu/lib/chea.jsp, accessed on August 6, 2015). We tested for enrichments of the target genes identified by ChIP for each of these 52 TFs to predicted TFBSs from our model.

**Enrichments of TF-target gene modules for gene ontology terms**

We downloaded Gene Ontology (GO) annotations for mouse genes from GO.db on November 4, 2015, using the topGO R package. We extracted the genes annotated to each GO term and its children, and we used Fisher's exact tests to characterize enrichments of TF-target gene modules for the 4,624 GO terms that contain between 10 and 500 genes.

**Enrichments of TF-target gene modules for cell-type-specific genes**

We characterized sets of genes expressed in each striatal cell type using gene expression profiles from purified cell types (Doyle *et al*, 2008; Zhang *et al*, 2014) and the pAppendix R package for cell-type-specific expression analysis (Dougherty *et al*, 2010). We used Fisher's exact tests to characterize enrichments of TF-target gene modules for genes expressed specifically in each cell type.

**Enrichments of TF-target gene modules for differentially expressed genes**

We identified genes that were differentially expressed in HD versus control samples. In the primary dataset, we compared mice with the non-pathogenic Q20 allele and mice with each of the other five alleles, separately for 2-, 6-, and 10-month-old mice. We used the edgeR R package to fit generalized linear models and test for significance of each contrast. We used Fisher's exact tests to characterize enrichments of downregulated genes and upregulated genes in each condition (significance threshold for differentially expressed genes, $P < 0.01$) for the target genes of each TF. We considered enrichments to be statistically significant at a raw *P*-value threshold < 1e-6, or an adjusted *P*-value < 0.02 after accounting for 19,170 tests (639 TFs × 5 *Htt* alleles × 3 time points × 2 tests/condition).

To identify top TFs, accounting for non-independence among genes and conditions, we calculated an empirical false discovery rate for these enrichments. We repeated the edgeR and enrichment analyses 1,000 times with permuted sample labels. We found that no module had a *P*-value < 1e-6 in more than four conditions in any of the permuted datasets. Therefore, we focused on TFs whose target genes were overrepresented for differentially expressed genes in five or more conditions.

We performed similar analyses to characterize TF-target gene modules enriched for genes that were differentially expressed in replication samples. We used the limma R package to calculate differentially expressed genes in each of the four microarray studies from mouse striatum (Kuhn *et al*, 2007; Becanovic *et al*, 2010; Fossale *et al*, 2011; Giles *et al*, 2012). We calculated enrichments of the DEGs from each study for TF-target gene modules. We then combined the enrichment *P*-values across the four studies using Fisher's method to produce a meta-analysis *P*-value for the association of each TF-target gene module in HD mouse models.

We used quantitative proteomics data from 6-month-old $Htt^{Q20/+}$, $Htt^{Q80/+}$, $Htt^{Q92/+}$, $Htt^{Q111/+}$, $Htt^{Q140/+}$, and $Htt^{Q175/+}$ mice ($n = 8$ per group) (Langfelder *et al*, 2016). We characterized proteins whose abundance was correlated with *Htt* CAG length in the striatum of 6-month-old mice, using MaxQuant protein quantities. We then calculated enrichments of CAG length-correlated proteins (Pearson's correlation, $P < 0.01$) for each TF-target gene module with Fisher's exact test, separately for proteins whose abundance was positively or negatively correlated with CAG length.

We used the limma R package to fit a linear model to characterize differentially expressed genes in each of two microarray datasets (Hodges *et al*, 2006; Durrenberger *et al*, 2015) profiling dorsal striatum of HD cases versus controls, treating sex as a covariate. We calculated enrichments of the DEGs from each study for TF-target gene modules. We then combined the enrichment *P*-values across the two studies using Fisher's method to produce a meta-analysis *P*-value for the association of each TF-target gene module with HD.

### Mouse breeding, genotyping, and microdissection

The B6.$Htt^{Q111/+mice}$ (Strain 003456; JAX) used for the ChIP-seq study have a targeted mutation replacing a portion of mouse *Htt* (formerly *Hdh*) exon 1 with the corresponding portion of human *HTT* (formerly *IT15*) exon 1, including an expanded CAG tract (originally 109 repeats). Mice used in the present study were on the C57BL/6J inbred strain background (Langfelder *et al*, 2016; Ament *et al*, 2017). Cohorts of heterozygote and wild-type littermate mice were generated by crossing B6.$Htt^{Q111/+}$ and B6.$Htt^{+/+}$ mice. Male mice were sacrificed at $122 \pm 2$ days of age (or 16 weeks) and 11 months via a sodium pentobarbital-based euthanasia solution (Fatal Plus, Henry Schein). Both hemispheres of each animal's brain were microdissected on ice into striatum, cortex, and remaining brain regions. These tissues were snap-frozen and stored in −80°C. Experiments were approved by an institutional review board in accordance with NIH animal care guidelines.

### Western blot

Male and female $Htt^{Q111/+}$ and wild-type littermates at 4 and 11 months of age were euthanized with sodium pentobarbital and

brains microdissected as described above. Striatal tissue was disrupted and homogenized in lysis buffer (Cell Signaling Technology, #9803) containing protease and phosphatase inhibitors (Thermo, #78443) using a syringe and 26-ga needle and then sonicated twice for five-seconds on ice. Debris was pelleted by centrifuging for 20 min at 13,000 *g* assay. Protein concentration was determined by BCA assay (Thermo, #PI23225), and 50 µg of denatured protein was prepared in LDS sample buffer (Invitrogen, NP0008). For quantitative Western blot analysis, the experimenter was blinded to both genotype and age and the protein was loaded in randomized order then run on 10% bis-tris polyacrylamide gels with MOPS running buffer (Invitrogen, NP0004, NP0001, and NP0302). Protein was transferred to low-fluorescence PVDF membranes (Immobilon-FL; Millipore) and total protein quantified for loading normalization (LiCor, #926-11010; LiCor Odyssey Fc Imager). All membrane wash steps were performed in tris-buffered saline with 0.05% Tween-20. Membranes were blocked (LiCor #927-50100) for 45 min before incubation in primary antibody against phospho-SMAD3 (Abcam ab52903; 1:500, 72 h at 4°C) and total SMAD3 (Invitrogen #MA5-15663; 1:500, 72 h at 4°C) prepared in the blocking solution with 0.05% Tween-20. Secondary antibodies used were goat anti-rabbit and goat anti-mouse (LiCor #925-32210, #925-32211, #925-68070, and #925-68071; 1:150,000) made in blocking buffer with 0.05% Tween-20 and 0.01% SDS. Quantitation of signal was performed using Image Studio v5.2 (LiCor) with the experimenter remaining blinded to genotype and age. SMAD3 signal was normalized to total protein stain.

### High-resolution X-ChIP-seq

We prepared duplicate ChIP samples for each antibody from 4-month-old $Htt^{Q111/+}$ and from age-matched wild-type mice. For each ChIP preparation, chromatin DNA was prepared using the combined striatal tissue from both hemispheres of three mice. Preliminary experiments suggested that this was the minimal amount of material required to provide enough material for multiple IPs. Striata were transferred to a glass dounce on ice and homogenized in cold PBS with protease inhibitors. High-resolution X-ChIP-seq was performed as described (Skene *et al*, 2010), with slight modifications. IPs were performed using Abcam anti-SMAD3 antibody ab28379 [ChIP grade] or anti-RNA polymerase II CTD repeat YSPTSPS antibody [8WG16] [ChIP Grade] ab817. Sequencing libraries were prepared from the isolated ChIP DNA and from input DNA controls as previously described (Orsi *et al*, 2015). Libraries were sequenced on an Illumina HiSeq 2500 sequencer to a depth of ~17–25 million paired-end 25-bp reads per sample. Sequence reads have been deposited in GEO, accession GSE88775.

### ChIP-seq analysis

Sequencing reads were aligned to the mouse genome (mm9) using bowtie2 (Langmead & Salzberg, 2012). Peak-calling on each sample was performed with MACS v2.1 (Zhang *et al*, 2008), scaling each library to the size of the input DNA sequence library to improve comparability between samples. We retained peak regions with a significant MACS *P*-value (FDR < 0.01 and a read count ≥ 10 in at least two of the individual ChIP samples). Enrichment of the SMAD3 motif (JASPAR CORE MA0513.1) was performed with CentriMo

(Bailey & Machanick, 2012), using the 250-bp regions around peak summits obtained by running MACS on the combined reads from all the samples. Peaks were mapped to genes using the chipenrich R package (Welch *et al*, 2014), and genes were ranked by the number of peaks within 10 kb of each gene's transcription start sites. Gene Ontology enrichment analysis of the top SMAD3 target genes (peak counts > 2 SD above the mean) was performed using Fisher's exact test, using the same set of GO terms used to analyze the computationally derived TF-target gene modules. Statistical analysis of differential occupancy in $Htt^{Q111/+}$ versus wild-type mice was performed with edgeR (Robinson *et al*, 2010).

### Software and primary data resources

Code for analysis of gene expression, transcriptional regulatory networks, and ChIP-seq data for this manuscript are publicly available in the github repository located at https://github.com/seth-ament/hd-trn. BedGraph files and raw sequencing data for SMAD3 and RNA Pol2 ChIP-seq can be accessed at the GEO repository GSE88775.

**Expanded View** for this article is available online.

## Acknowledgements
This work was supported in part by a contract from the CHDI Foundation (N.D.P., Principal Investigator, and J.B.C, Principal Investigator) and with internal funds from the Institute for Systems Biology. J.R.P. was supported by a National Science Foundation Graduate Research Fellowship. D.E.B. was supported by a Donald A. King summer fellowship from the Huntington's Disease Society of America, and J.B.C. was supported by Huntington Society of Canada New Pathways Program.

## Author contributions
SAA, JRP, JBC, and NDP designed research. SAA performed the computational analysis and built the TRN model. JPC, RMB, and SRC performed mouse work and Western blots. JRP and DEB conducted the ChIP-seq experiments with assistance from PJS. SAA and JRP wrote the manuscript. All authors including VCW, MEM, NSB, JR and LEH contributed to editing and revising the paper.

## Conflict of interest
The authors declare that they have no conflict of interest.

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
