## [Review Process File · Molecular Systems Biology]

Transcriptional regulatory networks underlying gene expression changes in Huntington's disease

Seth A. Ament, Jocelynn R. Pearl, Jeffrey P. Cantle, Robert M. Bragg, Peter J. Skene, Sydney R. Coffey, Dani E. Bergey, Vanessa C. Wheeler, Marcy E. MacDonald, Nitin S. Baliga, Jim Rosinski, Leroy E. Hood, Jeffrey B. Carroll and Nathan D. Price

Review timeline:

Submission date:	10 November 2016
Editorial Decision:	23 January 2017
Revision received:	13 December 2017
Editorial Decision:	15 January 2018
Revision received:	23 January 2018
Accepted:	26 February 2018

Editor: Thomas Lemberger

Transaction Report:

1st Editorial Decision

23 January 2017

Thank you again for submitting your work to Molecular Systems Biology. We have now heard back from the three referees who agreed to evaluate your manuscript. First of all, I would like to apologize for the delay in getting back to you, which was due to the late arrival of the report from referees after the Christmas break. As you will see from the reports below, the referees find the topic of your study of potential interest. They raise, however, important concerns on your work, which should be addressed in a revision.

The main concerns are expressed by referee #3 and refer to the need more support for the biological relevance of some of the key results. We appreciate that you use separate functional genomics and proteomics datasets to replicate the prioritization of the core set of TFs and that a TF-TF regulatory network is inferred. The biological and functional significance of the inferred regulatory relationships displayed in this network remains however rather speculative at this stage and its relevance to design potential protective strategies is not clear. With regard to SMAD3, we appreciate that the chromatin location study is consistent with a down-regulation of SMAD3-regulated transcription in HD mouse models and with the previously protective role of TGFbeta in vitro (Ring et al 2015). Whether TGF/SMAD signaling is down-regulated in medium spiny neurons in HD mouse striata remains however unclear. In view of the comments of reviewer #3 asking for stronger validation, it seems that some advance on one or the other of these two major open issues could considerably strengthen the manuscript.

In terms of presentation, not sufficient details are provided to understand and reproduce the analysis. Thus it is unclear how the TF-TF network was actually inferred. Overlap between gene sets (differentially expressed, sets of target genes, bound genes) are reported in terms of statistical

significance but the size of the overlap ('effect size') is often not reported, making it very difficult to intuitively understand the potential biological relevance of some of the results.

When you resubmit your manuscript, please download our CHECKLIST (<http://embopress.org/sites/default/files/Resources/EP_Author_Checklist_Master.xlsx>) and include the completed form in your submission. *Please note* that the Author Checklist will be published alongside the paper as part of the transparent process <<http://msb.embopress.org/authorguide#transparentprocess>>.

If you feel you can satisfactorily deal with these points and those listed by the referees, you may wish to submit a revised version of your manuscript. Please attach a covering letter giving details of the way in which you have handled each of the points raised by the referees. A revised manuscript will be once again subject to review and you probably understand that we can give you no guarantee at this stage that the eventual outcome will be favorable.

REVIEWER REPORTS

Reviewer #1:

The manuscript "Genome-scale transcriptional regulatory network 1 models for the mouse and human striatum predict roles for SMAD3 and other transcription factors in Huntington's disease" is well written and very informative about the identification of transcriptional factors that play a role in HD. The study provides a mouse and human striatal-specific TRN and prioritizes a hierarchy of transcription factor drivers in HD. The work is outstanding and the following up on SMAD3 compelling. The authors are very clear about the precision of the methods. Some questions that remain on the analysis are the following:

- (1) Did the authors find transcription factors that regulate BDNF levels in their analysis? They mention other publications but did not discuss this. Also were the transcription factors identified overlapping with the BDNF knockout mice transcriptional changes in their study?
- (2) I think cortex TF analysis is highly relevant and should be included as a comparison.
- (3) Are the changes in SMAD3 relevant to a particular neuronal type in the brain? Did they compare their analysis to single cell transcriptional changes in HD?

Minor points

- (1) Maybe change wording on (However, to our knowledge a role for SMAD3 has not been described). "Consistent with our work Ring et al identified but did not characterize transcriptional changes in SMAD3. See "Targets of the transcription factor SMAD3 were overrepresented ($p = 0.0018$), and the top WikiPathway was BDNF signaling ($p = 0.0053$)." Ring et al.
- (2) Reference line 30 Zuccato et al 2007 is in different font
- (3) Font on figures 3 and 4 should be larger

Reviewer #2:

This paper seems to be very mature, so perhaps I am seeing it after some rounds of revision. The method is a bit complex and not cutting-edge novel, but it is an interesting and novel use of TF-networks to investigate HD. Recent work has shown that CAG repeats yield tremendous phenotypic (and probably endotypic) complexity (Alexandrov et al. Nature Biotech 2016), so this paper is a very valuable resource for starting to tease apart the underlying molecular complexity of Htt repeat mutations. I applaud that the authors have released all of their analysis code in to the public domain via GitHub. If I were to be nit picky I think that Figure 1a gets a bit confusing where they go from linear models to networks as they don't really use the networks per se, though they can use the results of the linear models to paint networks. However, the paper is really about using data

integration to rank TFs and their targets. That being said it's a very minor comment. I think this is a well written paper with important results and clear and detailed analysis and is suitable for publication in MSB.

Reviewer #3:

Summary

Ament et al. reconstructed a genome scale transcriptional regulatory network (TRN) Huntington disease (HD) model. They did this by using digital genomic footprinting to predict the binding site of 871 TFs and tested the accuracy by comparing the TFBS with ChIP-seq data. They looked at co-expression patterns in RNA-seq transcriptome profiles to identify TF-target gene interactions in the mouse striatum and validated the predictive accuracy of the model by comparing the prediction with the observed expression level of each gene.

To identify TFs that are core regulators of transcriptional changes in HD they used published RNA-seq data from mice expressing different alleles of the HTT pathogenic variants. They identified a 48 TF core set whose target genes were differentially expressed (DEG) in 2, 6 and 10 months old mice. Published microarray and proteomics data sets were used to confirm the prediction for the selected 48 TFs core sets. In addition, the authors tested whether disease target genes identified in mouse, were differentially expressed in human late stage HD. They found a subset of 13 TFs of the initial 48 TFs differentially expressed in the mouse RNA-seq, mouse microarray, mouse proteomics and published human microarray data. Finally, the authors selected one TF, SMAD3, and performed ChIP-Seq experiments with 4 months old HttQ111 and wildtype striatum samples and found many SMAD3 targets downregulated.

Altogether, the authors aim in providing a global model between HD-related transcriptomic changes mediated by core TFs including a mouse and human striatal-specific TRN and a hierarchy of TFs.

General remarks

HD is a devastating disease and the authors merged and integrated published HD datasets of mice and men to extract TRNs that might trigger downstream pathological processes. Conceptually, this integration of published data into a TRN model, can be valuable to identify pathogenic molecular mechanisms that if known, might be drug targets resulting in neuroprotection. This HD TRN model can be a primer for biomedical research. The genome-scale TRN analysis sounds technically solid, but a more detailed explanation within the text and the method section is needed. However, it remains unclear whether the identified core TFs including SMAD3, have a biological impact in HD pathology or are "bystanders", since none of the identified core TFs have been tested for HTT interactions.

The in silico TRN analysis alone is interesting as a model, but without experimental validation, it remains elusive if the findings have a biological significance.

Major points

My major concerns are that it is unclear how biologically relevant is the TRN analysis, since none of the 13 core TFs have been experimentally validated in HD neither by the authors, nor by other researchers. The discussion is very short and superficial and a great opportunity to position the TRN analysis within the HD field is unused. Furthermore, taking advantage of published datasets has the downside that each set comprises different mouse models, ages, sample preparations and other methods, which might impede the analysis and must be better addressed in the manuscript.

1. The authors claim in the abstract to provide a "hierarchy of transcription factor drivers in HD". Where is this addressed in the manuscript? It is unclear.
2. Page 4 (lines 11-13): Since none of the TF drivers have been experimentally validated one should phrase "potential transcription factor drivers..." Otherwise, it is overstated.
3. Pages 6 and 7, the core TFs in independent datasets are not presented and discussed properly, for example only 22 of 48 TFs were enriched in protein samples and 13 of 48 core TFs were found in human samples. Please discuss this low fractions whether it stems from the analysis or from the samples (late human HD versus early onset mouse HD, ...).

4. Page 7 line 12 "This overlap was not statistically greater than expected by chance" and then the parameters were adjusted to obtain statistically significant differences? It sounds that the human data doesn't correlate with the mouse data and the conclusions are overstated (lines 19 -22).
5. Why was SMAD3 selected? Figure 3B does not support SMAD3 to be a top candidate in human HD.
6. Why were four-month-old HttQ111 mice used and not 2, 6 or 10 months old for SMAD3 ChIP-seq? This is inconsistent and a repetition with age-matched samples is needed.
7. Why were HttQ111 mice chosen and not Q140 or Q175 that have more DEGs at 6 and 10 months?
8. What is the biological and functional link between SMAD3 and the other core TFs with mutated HTT protein? An experimental validation for some of the core TFs is needed to support the in silico model.

Minor points

1. The TRN model (pages 4-5) needs a comprehensive description for a broader audience.
2. The mouse models used (page 6) need better explanation; for example when does the neurodegeneration start. It is painful to look up multiple references to obtain an overview.
3. How old were the mice used for microarray data (page 6)?
4. The TF-to-TF regulatory interaction analysis is not sufficiently explained (page 7). Figure 4 is also unreadable; one should consider a comprehensive graphical presentation.
5. Why does the peak range to a TSS vary among analysis, initially set to 5kb and for SMAD3 it is 10kb. This is obviously inconsistent.
6. Overall, a more detailed methods section would strengthen the manuscript.
7. Figure 2 is better suited for a supplementary figure. One could also consider adapting the color code to the other figures.
8. Overall, the readability and clarity could be improved to invite readers from the biomedical field.

To: Thomas Lemberger, PhD, Chief Editor
Molecular Systems Biology

Re: MSB-16-7435
December 13, 2017

Dear Dr. Lemberger,

Thank you for providing us the opportunity to submit a revised version of our manuscript, "Transcriptional regulatory networks underlying gene expression changes in Huntington's disease." In response to your suggestions and those of the peer reviewers, we have performed new experiments assessing age- and *Htt* CAG length-dependent changes in the expression of SMAD3 transcripts and proteins. We also added new analyses and substantially revised the text for clarity.

The revision process took eight months to complete, and we apologize for the slow timeframe. Both lead authors on the manuscript, Drs. Ament and Pearl, transitioned to new positions over the last several months. Dr. Ament is now an Assistant Professor at the University of Maryland School of Medicine. Dr. Pearl is now a Fellow at the Altius Institute for Biomedical Sciences. These transitions led to inevitable delays in completing new experiments and analyses and in revising the manuscript. Under the circumstances, I hope you will still consider this as a revision rather than as a new submission.

Below is a summary of the changes we have made in response to specific referee comments.

Editorial Suggestions.

At your suggestion, we have expanded our methods section to clearly describe our analyses so that other groups are able to reproduce our work.

Another main concern was the need for additional biological relevance of some of the key results. In order to address this, we performed additional experiments to examine changes in SMAD3 and phosphorylated-SMAD3 in 4 and 11-month-old mice. On page 8, we added several sentences framing the literature and results around SMAD3 to better orient the reader to the importance of our results and others exploring the role of this transcription factor in HD.

Reviewer 1.

- (1) Did the authors find transcription factors that regulate BDNF levels in their analysis? They mention other publications but did not discuss this. Also were the transcription factors identified overlapping with the BDNF knockout mice transcriptional changes in their study?
- (2) I think cortex TF analysis is highly relevant and should be included as a comparison.
- (3) Are the changes in SMAD3 relevant to a particular neuronal type in the brain? Did they compare their analysis to single cell transcriptional changes in HD?

We thank the reviewer for their helpful feedback and questions. In response to (1) we have expanded the results section "Biological associations of core TFs" on page 7. In response to point (2) we have added a paragraph on page 6 lines 27-41 which discusses gene expression changes in four additional tissues (cortex,

liver, hippocampus, and cerebellum). This analysis reveals a range of tissue specificity within the predicted TF modules. For example, predicted gene targets of RXRG were enriched in all five tissues, whereas predicted targets of IRF2 were enriched only in striatum. The results of this analysis are described in supplemental Figure 7. In response to (3), we were certainly interested in the relevance of particular TFs to neuronal or other cellular subtypes in the brain. We include in the lower panel of Figure 5 the cell-type specific enrichments of each TF's target genes across seven major cell types in the brain. We observe a predominant enrichment for these genes in Drd1 or Drd2+ neurons. We welcome the reviewer's question about single-cell transcriptional changes in HD. At this time, we are not aware of any single-cell transcriptomic data related to HD that is available to us. Data of this kind would be particularly helpful in understanding the contributions of particular cell types to gene expression changes observed in bulk tissues such as striatum.

Reviewer 2.

This paper seems to be very mature, so perhaps I am seeing it after some rounds of revision. The method is a bit complex and not cutting-edge novel, but it is an interesting and novel use of TF-networks to investigate HD. Recent work has shown that CAG repeats yield tremendous phenotypic (and probably endotypic) complexity (Alexandrov et al. Nature Biotech 2016), so this paper is a very valuable resource for starting to tease apart the underlying molecular complexity of Htt repeat mutations. I applaud that the authors have released all of their analysis code in to the public domain via GitHub. If I were to be nit picky I think that Figure 1a gets a bit confusing where they go from linear models to networks as they don't really use the networks per se, though they can use the results of the linear models to paint networks. However, the paper is really about using data integration to rank TFs and their targets. That being said it's a very minor comment. I think this is a well written paper with important results and clear and detailed analysis and is suitable for publication in MSB.

We thank the reviewer for their positive feedback.

Reviewer 3.

Major points

My major concerns are that it is unclear how biologically relevant is the TRN analysis, since none of the 13 core TFs have been experimentally validated in HD neither by the authors, nor by other researchers. The discussion is very short and superficial and a great opportunity to position the TRN analysis within the HD field is unused. Furthermore, taking advantage of published datasets has the downside that each set comprises different mouse models, ages, sample preparations and other methods, which might impede the analysis and must be better addressed in the manuscript.

We have added several new experiments and analyses to address these concerns. Focusing on SMAD3, we conducted new experiments characterizing the effects of age and HTT CAG length on SMAD3 transcript and protein levels. Together with our ChIP-seq experiment confirming an association between SMAD3 binding and HTT CAG-length dependent gene expression changes, as well as previous work linking the SMAD3 agonist TGF-beta to Huntington's disease, these data provide a compelling case for a biological connection of SMAD3 signaling in HD. In the discussion (page 10-11) we highlight literature that supports roles in HD for several TFs that were found to be core TFs in our analysis, including RARB, FOXO1, KLF16, and RXRG. In response to the reviewer's final comment about differences in mouse models, ages etc. we have added an additional sentence in the discussion (page 10, lines 40-43). In our view, observing enrichments for differential expression for the same TFs across multiple time points and mouse models suggests that the findings are robust. However, we agree with the reviewer that a more refined view of network dynamics will require careful attention to the mouse models being used.

1. The authors claim in the abstract to provide a "hierarchy of transcription factor drivers in HD". Where is this addressed in the manuscript? It is unclear.

We thank the reviewer for this point and have removed this phrase from the manuscript.

2. Page 4 (lines 11-13): Since none of the TF drivers have been experimentally validated one should phrase "potential transcription factor drivers..." Otherwise, it is overstated.

We have revised the language describing "potential transcription factor drivers" to "predictions" of transcription factor drivers throughout the manuscript.

3. Pages 6 and 7, the core TFs in independent datasets are not presented and discussed properly, for example only 22 of 48 TFs were enriched in protein samples and 13 of 48 core TFs were found in human samples. Please discuss this low fractions whether it stems from the analysis or from the samples (late human HD versus early onset mouse HD, ...).

We thank the reviewer for this comment. We have revised the section describing the datasets and hope that this has improved its clarity. For the comparison to four independent microarray datasets in mouse models of HD, we found that targets of 46 of the 48 core TFs were enriched for DEGs, an overlap which is significantly greater than expected by chance (Fisher's exact test: $p = 5.7e-32$). We also found in a comparison with protein quantity data that targets of 22 of the 48 core TFs were enriched for differentially abundant proteins, an overlap which is significantly greater than expected by chance (Fisher's exact test: $p = 5.7e-20$). We revised the paragraph describing the comparison to late-stage human HD and address that specific comment below.

4. Page 7 line 12 "This overlap was not statistically greater than expected by chance" and then the parameters were adjusted to obtain statistically significant differences? It sounds that the human data doesn't correlate with the mouse data and the conclusions are overstated (lines 19 -22).

We thank the reviewer for this point. We chose to perform two tests in which we considered either a restrictive set of TFs from the HD mouse models (the 48 core regulators), as well as a broader set of TFs (all 209 TFs whose predicted target genes were enriched in at least one condition from our primary mouse RNA-seq dataset). We state that for the overlap of 13 of 48 TFs, a p-value of 0.05 is not statistically significant (page 6, lines 9-10). We also reiterate that we are comparing advanced or late stage human disease to early pathogenic states in mouse models.

5. Why was SMAD3 selected? Figure 3B does not support SMAD3 to be a top candidate in human HD.

We address the selection of SMAD3 on page 8, lines 9-15. Briefly, SMAD3 was predicted to be a driver of gene expression changes across 4 independent datasets, and SMAD3 targets were predominantly down-regulated in an age- and CAG length-dependent manner beginning relatively early. In our view, samples from late-stage human HD are not useful for ranking top candidates as drivers of disease progression, since human samples include only the tail end of the disease progression. The extensive cell death and other gross pathology make it difficult to interpret the changes in the human samples. We, like many others in the HD research community, are therefore most focused on identifying early drivers, which can be detected most directly in the mouse models.

6. Why were four-month-old HttQ111 mice used and not 2, 6 or 10 months old for SMAD3 CHIP-seq? This is inconsistent and a repetition with age-matched samples is needed.

We utilized 4 month old mice because we wanted to capture early changes in SMAD3 binding. This decision was made based both on the saturation of differential gene expression changes observed at six months in the allelic series study (Langfelder et al), and also based on our knowledge of gene expression changes in a dense time series study of Htt^{Q111/+} mice that was recently published in Human Molecular Genetics (Ament et al 2017). Specifically, we were interested in capturing a time point at which the down-regulation of neuronal genes has begun – these are the gene expression changes that SMAD3 is predicted to regulate -- but prior to the onset of neuroinflammatory gene expression changes that occur slightly later. By six months, neuroinflammatory gene expression changes are apparent in the Q111, Q140, and Q175 mouse models.

7. Why were HttQ111 mice chosen and not Q140 or Q175 that have more DEGs at 6 and 10 months?

A motivating factor of our study was to identify very early changes in the transcriptome that are also representative of human disease. For this reason, we prefer to use the shortest CAG-repeat length that allows us to detect early differences (Htt^{Q111/+} mouse model). Additionally, many phenotypic studies are based in the Q111 mouse model, making it easier to understand transcriptional changes as they relate to other phenotypes.

8. What is the biological and functional link between SMAD3 and the other core TFs with mutated HTT protein? An experimental validation for some of the core TFs is needed to support the in silico model.

We thank the reviewer for this question. We agree that understanding the biological and functional link between the core TFs and mutated HTT protein is a promising future direction. On page 10, lines 32-43, we frame the limitations of our study, and suggest that the replication of 13 core TF predictions across four independent datasets sets up a clear set of hypotheses for follow-up studies. In an effort to further understand the protein-level changes of SMAD3, we looked at protein abundance differences in SMAD3 and phosphophorylated-SMAD3 in the striatum of Htt^{Q111/+} mice compared to controls. This is included as an additional supplemental figure (page 33).

Thank you very much for your consideration.

Sincerely,

Nathan Price

Nathan Price
Professor & Associate Director
Institute for Systems Biology
Seattle, WA
401 Terry Ave N
Seattle, WA 98109

(206) 732-1204

nprice@systemsbiology.org

Thank you again for submitting your work to Molecular Systems Biology. We are now globally satisfied with the modifications made and we will be able to accept your paper for publication in Molecular Systems Biology pending the following minor modifications:

Figures

- The figure files are still too small. Please change the resolution to 300PPI/inch *at the final size of the image* (please see http://www.embopress.org/sites/default/files/EMBOPress_Figure_Guidelines_061115.pdf)
- The figures should only appear once, please remove the duplicates from the ms file.
- Figure 2 does not need to be named 2A since there is only one panel.
- For figure panels showing network visualization, we would encourage you to supply the corresponding zipped Cytoscape .cys files as "Figure Source Data Files" such that readers can download these files directly from the figures.

Callouts

- All the main figure panels should be called out. Please add the missing ones: Fig 1E, Fig 3A-B, Fig 4A-D, Fig 5A-D, fig 6G-I.
- If possible, it would be better to call out Appendix figures from the main text. Please add callouts to Appendix figure S5 and S8.
- The callout to Appendix fig S2 appears before S1 (ideally they should be in numerical order)
- There is a callout to Appendix Dataset 1 on page 4, line 27, to Appendix Dataset 2 on page 5, line 22 and to Dataset 3 on page 9, line 8. Please update all Dataset callouts to "Dataset EV1-3" and please name the corresponding files Dataset EV1-3 (also see below).

Datasets

- Mouse Striatum TRN Edges, TF to target gene module enrichments for reference datasets, TF module enrichment across 5 tissues: Please rename them Dataset EV1-3, add legends in separate tabs and update the callouts in the text.

Supplementary information

- Please rename the figures and tables from 'SI Figure/Table [n]' to 'Appendix Fig. /Table S[n]'. Apologies for being picky, but The "S" for Fig. Sn is still missing, both in the Appendix and the callouts in the main ms.

Thank you for tentatively accepted our manuscript, "Transcriptional regulatory networks underlying gene expression changes in Huntington's disease" pending the following minor modifications. Listed below are the amendments we have made to the text and figures you requested.

Figures

- The figure files are still too small. Please change the resolution to 300PPI/inch *at the final size of the image* (please see http://www.embopress.org/sites/default/files/EMBOPress_Figure_Guidelines_061115.pdf)

We have increased the resolution of the figures and uploaded new versions.

- The figures should only appear once, please remove the duplicates from the ms file.

We have removed the figures from the manuscript text file.

- Figure 2 does not need to be named 2A since there is only one panel.

We removed the '2A' label and updated this figure to a higher resolution version.

- For figure panels showing network visualization, we would encourage you to supply the corresponding zipped Cytoscape .cys files as "Figure Source Data Files" such that readers can download these files directly from the figures.

We have included the .cys file.

Callouts

- All the main figure panels should be called out. Please add the missing ones: Fig 1E, Fig 3A-B, Fig 4A-D, Fig 5A-D, fig 6G-I.

We have added callouts in the manuscript text for Fig 1E (page 4, line 27), Fig 3A-B (page 5 line 29 and 39), Fig4A-D (page 7 line 7), and Fig 6G-I (Page 9 line 40). We amended the description of Fig 5, and it is called out simply as Fig 5 in the text.

- If possible, it would be better to call out Appendix figures from the main text. Please add callouts to Appendix figure S5 and S8.

We have added callouts for S5 on page 5 line 20 and S8 on page 8 line 30 and 32.

- The callout to Appendix fig S2 appears before S1 (ideally they should be in numerical order)

We have reordered these two figures.

- There is a callout to Appendix Dataset 1 on page 4, line 27, to Appendix Dataset 2 on page 5, line 22 and to Dataset 3 on page 9, line 8. Please update all Dataset callouts to "Dataset EV1-3" and please name the corresponding files Dataset EV1-3 (also see below).

We have renamed the datasets and updated the callouts from (for example) "Appendix Dataset 2" to Dataset EV2.

Datasets

- Mouse Striatum TRN Edges, TF to target gene module enrichments for reference datasets, TF module enrichment across 5 tissues: Please rename them Dataset EV1-3, add legends in separate tabs and update the callouts in the text.

We have renamed the three datasets and updated the callouts in the text.

Supplementary information

- Please rename the figures and tables from 'SI Figure/Table [n]' to 'Appendix Fig. /Table S[n]'. Apologies for being picky, but The "S" for Fig. Sn is still missing, both in the Appendix and the callouts in the main ms.

We have updated the Appendix (for example) from Figure 1 to Figure S1.

We have updated the callouts in the manuscript to match the appendix names.

We hope this completes the modifications necessary for the publication of our manuscript in Molecular Systems Biology. Thank you for working with us to improve the paper for publication.

Corresponding Author Name: Nathan D Price
Journal Submitted to: Molecular Systems Biology
Manuscript Number: MSB-16-7435